# Development and evaluation of AI model with deep learning for segmentation of extraocular muscles in thyroid eye disease

Yusuke Haruna[1]*, Mizuki Tagami[1], Ryo Kurosaki[2], Mizuho Nishio[2,3], Gen Kinari[1], Mami Tomita[1], Norihiko Misawa[1,4], Yoshihiro Muragaki[2], Shigeru Honda[1]

1 Department of Ophthalmology and Visual Sciences, Graduate School of Medicine, Osaka Metropolitan University, Abeno-ku, Osaka-shi, Osaka-fu, Japan, 2 Department of Medical Device Engineering, Graduate School of Medicine, Kobe University, Chuo-ku, Kobe-shi, Hyogo Prefecture, Japan, 3 Departments of Radiology, Graduate School of Medicine, Kobe University, Chuo-ku, Kobe-shi, Hyogo Prefecture, Japan, 4 Department of Ophthalmology & Visual Sciences, Washington University School of Medicine, St. Louis, Missouri, United States of America

* mizuki1979feb@yahoo.co.jp

## Abstract

### Purpose

To develop and evaluate an AI model for the segmentation of extraocular muscles (EOMs) using Magnetic Resonance Imaging (MRI).

### Design

Single-center study with retrospective study.

### Methods

The study included 52 patients with thyroid eye disease (TED) who underwent MRI examination of the orbital region at the Department of Ophthalmology, Osaka Metropolitan University, between October 2020 and June 2023. Manual labelling of all EOMs was performed on all slices. An AI model was created and compared with the manually labelled data using 5-fold cross validation on data sets of 12, 32, and 52 cases. The average signal intensity ratio (SIR) within the EOMs was measured. The primary outcome was the comparison of DICE similarity coefficients (DSC) as the agreement rate between AI's result and manual labelling amongst the three data sets (52 cases vs 32 cases vs 12 cases). The secondary outcome was the correlation of SIR between the AI and manual labelling.

### Results

A significant difference in DSC was observed between the 12-, 32-, and 52-case data sets for the inferior rectus muscle and medial rectus muscle. A significant correlation was observed between the AI and manual labelling for SIR in all EOMs.

**Data availability statement:** Author-generated code is on github: https://github.com/bowang-lab/U-Mamba.

**Funding:** This work were supported by Japan Society for the Promotion of Science KAKENHI (Grant Number 23K09013(Mizuki Tagami),
+ 23KK0148(Mizuho Nishio). The funders had no role in study design, data collection and analysis, decision to publish, or preparation of the manuscript.

**Competing interests:** The authors have declared that no competing interests exist.

## Conclusions

An AI model was successfully developed for the automatic segmentation of EOMs. There was little measurement error between AI and manual labelling, and the AI model using 52 cases had improved measurement accuracy compared to the AI models using 32 and 12 cases.

## Introduction

Thyroid eye disease (TED) is an autoimmune disorder that primarily affects the orbit and periorbital tissues. It is characterized by inflammation and enlargement of the extraocular muscles (EOMs) and orbital fat, which can lead to various ocular symptoms including proptosis, diplopia, and vision loss [1,2]. Therefore, accurate evaluation of EOMs is crucial for quantifying disease activity, monitoring treatment response, and predicting outcomes in TED patients.

In recent years, Artificial Intelligence (AI) has seen significant advancements and has been increasingly applied to various areas within the medical field, particularly in medical imaging [3,4]. Magnetic Resonance Imaging (MRI) plays a vital role in the assessment of TED, offering detailed visualization of orbital structures and being useful for quantifying disease activity and predicting the response to anti-inflammatory treatment and the outcome in TED [5,6]. MRI is a powerful tool for evaluating EOMs involvement and disease activity in TED, as it can detect inflammatory edema before irreversible fibrosis occurs [7]. Despite the growing interest in AI applications in medicine [4], while there is some AI research using CT to diagnose and evaluate TED [8], AI research using MRI is still limited. CT is a useful test, but it is an invasive test that involves radiation exposure, whereas MRI noninvasively generates cross-sectional images of internal structures and allows for anatomical and inflammation assessment [9].

Currently, there is no clear standard for judging the presence or absence of activity in EOMs inflammation using MRI in the diagnosis and treatment of TED. To address this, we focused on the signal intensity ratio (SIR) in Short Tau/Time Inversion Recovery (STIR) sequence as an objective indicator for evaluating intraorbital inflammation. SIR on STIR or TIRM sequences has been reported as a promising objective biomarker that correlates with the Clinical Activity Score (CAS) and predicts treatment response [10]. Additionally, advanced techniques such as diffusion-weighted imaging have been explored to further objectify the 'tip of the iceberg' in orbital inflammation [11]. The primary objective of this study was to investigate the feasibility of developing an AI model capable of automatic segmentation of EOMs, which is a foundational step towards developing an AI model for evaluating the presence or absence of disease activity based on SIR. Contrast-enhanced T1-weighted images (T1WI) are the most accurate for volume segmentation, but assessing inflammation using only T1WI is difficult [5]. While previous studies have highlighted the value of combining multiple sequences, such as T1, T2, and STIR, for a comprehensive assessment of disease activity [7,10], our study focused on a single-sequence (STIR) AI model to

simplify the clinical workflow. This development of automated methods for segmenting and analyzing EOMs in TED could significantly improve diagnostic efficiency and precision.

## Methods

Prior to this study, we obtained approval from the Ethics Committees of the Osaka Metropolitan University (OMU) (approval no. 2023−072 First Approval: 26/09/2023, Second change approval: 31/03/2025, Third change approval:10/07/2025), Japan. All patients at OMU who participated in the study provided optout informed consent for their patient information to be stored in the hospital database and used in this study. This study was conducted in accordance with the tenets of the Declaration of Helsinki.

The study protocol was a single-center study and the protocol was as follows. This study included 52 patients diagnosed with moderate to severe TED who underwent plain MRI of the orbital region at the Department of Ophthalmology, OMU, between October 2020 and June 2023. The diagnosed severity was based on the European Thyroid Association/ European Group on Graves' Orbitopathy (EUGOGO) management guidelines [12]. All patients underwent MRI before treatment. MRI scans were performed using a coronal STIR sequence. MRI scans were performed using a 3.0T scanner [SIEMENS, MAGNETOM Vida] with a HeadNeck_20_TCS. STIR images were acquired in the coronal plane with the following parameters: repetition time (TR) = 4970ms or 5320ms or 5670ms, echo time (TE) = 59ms, and inversion time (TI) = 230ms. The slice thickness was 3 mm with a 0.3 mm interslice gap. The field of view was 180×180 mm, and the matrix size was 512×512. We accessed the medical records from 30/08/2024–20/12/2024 and exported them to a CD-R as a DICOM image series. Manual labelling of the EOMs—superior rectus muscle (SRM), inferior rectus muscle (IRM), medial rectus muscle (MRM), and lateral rectus muscle (LRM) —was performed on all identifiable slices of both eyes using the image analysis software ITK-snap. (Fig 1) ITK-snap is a segmentation tool widely used in radiology and oncology that allows for the creation of 3D models thru manual labelling [13]. Three ophthalmologist (YH, MT, and NM) performed the labelling. A board-certified radiologist (MN) visually confirmed the results of the labelling by the ophthalmologists. The average SIR within the EOMs was also measured. Additionally, signal intensity within the eyeball was measured by automatic segmentation of the eyeball using Hough transform. The SIR was defined as follows:

Signal intensity ratio (SIR)= EOM signal intensity /eyeball signal intensity

The segmentation model used was U-Mamba_Enc [14,15]. U-Mamba is a model that incorporates the Mamba architecture, which is an extension of State Space Models (SSMs), into nnU-Nett V2 [16,17]. nnU-Nett V2 is a semantic segmentation model specialized for medical images that is based on U-Nett, a Convolutional Neural Network (CNN)-based segmentation model, and has the ability to automatically adapt to a given dataset [18]. By using Selective SSMs, Mamba can capture long-distance dependencies that are difficult to capture with the local receptive field of CNN. Since U-Mamba

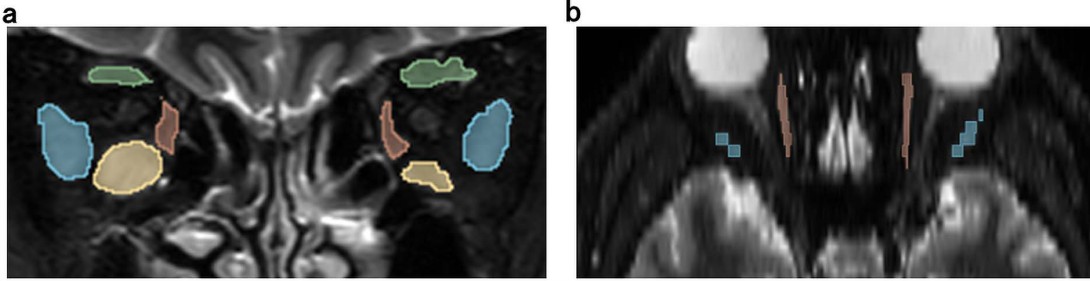

**Fig 1. Manual labeling EOMs on MRI (STIR sequence) using ITK-snap.** Labeling was done on the EOMs as shown in a (coronal image). b is a transverse images reconstructed from coronal images using ITK-snap. green: SRM, yellow: IRM, red: MRM, blue: LRM.

is a hybrid structure of CNN and Mamba, it was proposed as a method that can efficiently extract both local features and long-distance dependencies. There are two types of U-Mamba: U-Mamba_Enc, which uses U-Mamba blocks in all encoder blocks, and U-Mamba_Bot, which uses U-Mamba blocks only in the bottleneck. In this study, U-Mamba_Enc was used. The model input data was 3D STIR image data, and the ground truth labels were data that specified the area of the EOMs on STIR images at the pixel level. We used the default image preprocessing and default self-configurations that U-Mamba inherits from nnU-Nett V2. The performance of the final model was determined by the average score calculated by 5-fold cross-validation. We used the implementation of 5-fold cross-validation included in U-Mamba. An overview of 5-fold cross-validation is as follows. The input data for the model was divided equally into five folds. One fold was used for validation, and the remaining four folds were used for model training. Five patterns (5 splits) were prepared, with the test fold covering all folds. Regression model training and testing were performed for each split. The evaluation scores for all folds were averaged to determine the final evaluation score. (S1 Appendix).

The primary outcome was the comparison of the agreement rate between the AI model's results and manual labelling across the 12, 32, and 52-case datasets. The secondary outcome was to evaluate the correlation of SIR between the AI and manual groups across the 52-case datasets.

## Statistical analysis

All statistical analyses were performed with EZR (Version 1.60, Saitama Medical Centre, Jichi Medical University, Saitama, Japan), which is used for the calculation of R. More precisely, it is a modified version of R commander, which is designed to add statistical functions frequently used in biostatistics [19]. The Dice similarity coefficient (DSC) was used to quantify the accuracy of image segmentation as agreement rate between AI's results and manual labelling. One-way ANOVA and Tukey multiple comparisons test was employed for comparisons involving three or more groups. Spearman's rank correlation coefficient and Bland-Altman plots were used to assess the correlation between SIR measurements. Values of $P < 0.05$ were considered statistically significant.

## Results

An example of the AI's result is shown in Fig 1B. The DSC for each EOM is shown in Fig 2. For SRM, the DSC was $0.74 \pm 0.07$ in the 12 group, $0.75 \pm 0.06$ in the 32 group, and $0.75 \pm 0.06$ in the 52 group, with no significant differences observed between the groups. (One-way ANOVA: $p = 0.70$) For IRM, the DSC was $0.72 \pm 0.22$ in the 12 group, $0.82 \pm 0.07$ in the 32 group, and $0.82 \pm 0.06$ in the 52 group, with significant differences observed between the groups. (One-way ANOVA: $p = 0.011$) For MRM, the DSC was $0.78 \pm 0.08$ in the 12 group, $0.83 \pm 0.04$ in the 32 group, and $0.83 \pm 0.05$ in the 52 group, with significant differences observed between the groups. (One-way ANOVA: $p = 0.0092$) For LRM, the DSC was $0.69 \pm 0.22$ in the 12 group, $0.78 \pm 0.10$ in the 32 group, and $0.78 \pm 0.10$ in the 52 group, with no significant differences observed between the groups. (One-way ANOVA: $p = 0.087$) Turkey's multiple comparison test was performed on the IRM and MRM. For IRM, it found significant differences between 12 group and 32 group ($p = 0.020$) and between 12 group and 52 group ($p = 0.0094$). There was no significant difference between 32 group and 52 group ($p = 0.99$). For MRM, it found significant differences between 12 group and 32 group ($p = 0.0084$) and between 12 group and 52 group ($p = 0.015$). There was no significant difference between 32 group and 52 group ($p = 0.85$).

Fig 3 shows the Bland-Altman plots of SIR for the AI and manual measurements using data from 52 cases. The mean SIR for manual and AI measurements was $0.37 \pm 0.14$ for SRM, $0.36 \pm 0.15$ for IRM, $0.38 \pm 0.12$ for MRM, and $0.33 \pm 0.077$ for LRM. The difference in SIR between manual and AI measurements (manual-AI) was $-0.0080 \pm 0.020$ for SRM, $-0.012 \pm 0.017$ for IRM, $-0.0075 \pm 0.016$ for MRM, and $-0.016 \pm 0.013$ for LRM.

Fig 4 shows the correlation between AI and manually measured SIR using data from 52 cases. Spearman's correlation coefficients were 0.987 for SRM ($p = 0$), 0.97 for IRM ($p = 0$), 0.985 for MRM ($p = 0$), and 0.988 for LRM ($p = 0$), and significant strong correlations were observed for all EOMs.



**Fig 2. Concordance rate (DICE coefficient) between AI and manual EOMs labeling in each group (12 group vs 32 group vs 52 group).** Significant differences between groups were observed in IRM and MRM. a: SRM, b: IRM, c: MRM, d: LRM.

Fig 5 shows the representative case of AI model's result for segmentation of EOMs. Fig 1 and Fig 5(a,b) show the same slice from the same patient. Comparing Fig 1,5, there is a slight difference with SRM (Fig 5c,d: area surrounded by yellow oval). However, for the other EOMs, the AI' results were almost identical to the manual labelling. We found that the AI's result was almost as accurate as the manual labelling.

## Discussion

In this study, we successfully developed an AI model for the automatic segmentation of EOMs in patients with TED using MRI images. A significant difference in DSC was observed between the 12, 32, and 52-case groups for the IRM and MRM. This suggests that incorporating a larger sample size improved the performance of the AI model, particularly for these muscles. This finding is consistent with the general principle of deep learning that more data often leads to improved



Fig 3. Bland-Altman plots of SIR for the AI and manual measurements using data from 52 cases. The difference between the AI and Manual measurements was minor. a: SRM, b: IRM, c: MRM, d: LRM.

model performance [20]. The SRM showed no significant improvement in DSC with increased sample size. The lack of significant difference in the SRM may be attributed to the anatomical proximity and the shared fascial connections between the SRM and the levator palpebrae superioris muscle. In TED, these muscles are often collectively referred to as the Superior Rectus-Levator Palpebrae Superioris (SRLPS) complex due to their tendency to enlarge together and the frequent disappearance of the intervening fat plane on MRI. This anatomical characteristic likely led to partial volume effects and increased the difficulty of precise discrete segmentation, which may have affected the statistical results for the SRM. As mentioned in the introduction, using T1WI may result in more accurate segmentation. However, because MRI images cannot be captured under different conditions simultaneously, slight misalignments in muscle positions can easily occur between STIR and T1 images due to eye movements or slight head movements during imaging. Therefore, when analyzing T1 and STIR images together with a single AI model, "co-registration" is necessary to precisely overlay the two images at the pixel level. STIR images, while less accurate than T1 images, can be segmented, thus avoiding the risk of registration errors that occur in the process of combining multiple images. Furthermore, by loading only one image series,



**Fig 4. Spearman's correlation coefficients between AI and manually measured SIR using data from 52 cases.** Significant strong correlations were observed for all EOMs. a: SRM, b: IRM, c: MRM, d: LRM.

the clinical workflow can be simplified, making segmentation tools easier to use in clinical settings. Also, STIR images are superior for activity assessment in TED [5,21]. So, we decided to use STIR images for segmentation. In contrast, IRM and MRM are known to be the most commonly swollen EOMs in TED [22,23]. This distinct swelling may make them easier to differentiate from surrounding tissues, potentially leading to higher accuracy in manual labelling and, consequently, better training data for the AI model. DSC was 0.75 for SRM, 0.82 for IRM, 0.83 for MRM, 0.78 for LRM in an AI model using 52 cases. A similar study using TED computed tomography (CT) images reported a DSC of 0.902, which was higher than the present study [8]. This may be due to the large sample size of 281 and the fact that muscle tissue boundaries are less clear in the MRI (STIR sequence) compared to CT. In previous reports on AI-based automatic segmentation of the uterus

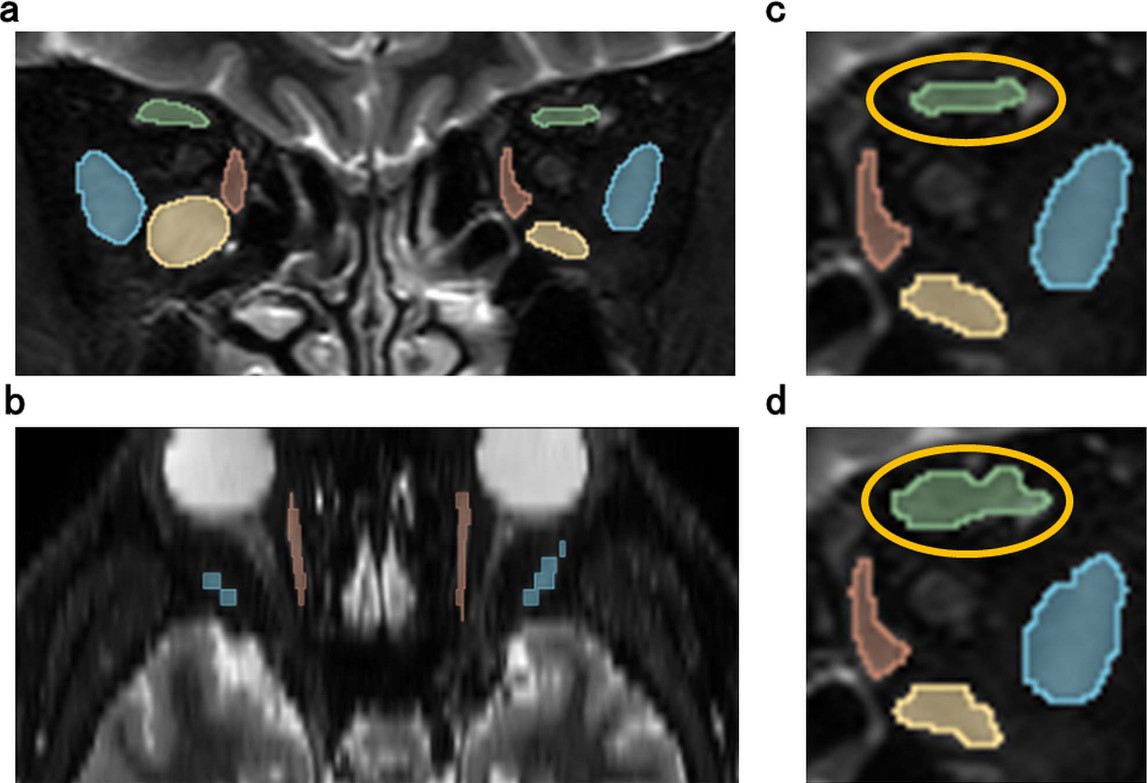

**Fig 5. AI's result of segmenting EOMs on MRI.** Fig 1 corresponds to Fig 5(a, b). C is an enlarged view of the left orbital area of Fig 5a, and d is an enlarged view of the left orbital area of Fig 1a. green: SRM, yellow: IRM, red: MRM, blue: LRM.

or prostate using T2-weighted MRI, the DSC was 0.927 [24], 0.793 [24], 0.82 [25], which was close to the results of this study.

Furthermore, a significant correlation was found between the AI group and the manual group in terms of the average SIR for all EOMs. The Bland-Altman analysis indicated minimal measurement errors between the two groups. While there are currently no established reference values for SIR, making it difficult to precisely evaluate the magnitude of our study's measurement error, previous reports comparing SIR before and after treatment in moderate to severe TED showed a decrease from $2.2 \pm 1.4$ to $1.8 \pm 1.4$ [26]. Given this background, it appears that the measurement errors observed in this study were minor and that the AI model is capable of accurately measuring the SIR of EOM. In this study, we evaluated the signal intensity of EOMs using the SIR, providing a quantitative basis for identifying 'hyperintensity' associated with active TED. While clinical scores (CAS) or healthy controls were not included in this dataset, the AI-derived SIR successfully represented a continuous spectrum of signal intensities. Higher SIR values extracted by the model correspond to the hyperintense areas typically indicative of active inflammation, as described in previous studies [10]. This demonstrates that our AI model can objectively 'inform' clinicians of muscle signal status without the inter-observer variability of visual inspection. Future research incorporating clinical activity data will be necessary to define the optimal SIR (for example, thresholds generally considered to be high signal intensity. normal signal intensity or hyperintensity) threshold for active TED. This study is the first step in objectively evaluating various MRI data in the future.

This study has several limitations. First, it was a single-center study. Therefore, it is necessary to verify whether the AI model developed in this study can be used on MRI at other facilities in the future. Second, we only included patients with



active TED. It remains to be verified whether the AI model can segment EOMs with similar performance in MRI without EOM swelling after treatment. Third, this segmentation is performed using only STIR images, so the volume estimate may differ slightly from the actual volume. We are trying to create an AI model that evaluates inflammation and volume using only one continuous image, so we chose STIR images, which allow assessment of both inflammation and volume.

## Conclusion

We successfully created an AI model capable of automatically segmenting EOMs from MRI images with the STIR sequence. This AI model has the potential to be a useful tool for real world TED practice in the future.

## Supporting information

**S1 Appendix. Five-fold cross-validation used in the current study.** The input data for the model was divided equally into five folds.(model 1–5) The evaluation scores for all folds were averaged to determine the final evaluation score. (DOCX)

## Author contributions

**Conceptualization:** Mizuki Tagami, Mizuho Nishio.

**Data curation:** Yusuke Haruna.

**Formal analysis:** Mizuki Tagami.

**Funding acquisition:** Mizuki Tagami.

**Investigation:** Yusuke Haruna, Mizuki Tagami, Ryo Kurosaki, Mizuho Nishio, Gen Kinari, Mami Tomita, Norihiko Misawa.

**Project administration:** Yusuke Haruna.

**Resources:** Yusuke Haruna.

**Software:** Yusuke Haruna, Mizuho Nishio.

**Supervision:** Mizuho Nishio, Norihiko Misawa, Yoshihiro Muragaki, Shigeru Honda.

**Writing – original draft:** Yusuke Haruna.

**Writing – review & editing:** Mizuki Tagami, Mizuho Nishio, Yoshihiro Muragaki, Shigeru Honda.

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
