## [Decision Letter · Decision Letter 0]

30 Mar 2026

PONE-D-26-06489Development and evaluation of AI model with deep learning for segmentation of extraocular muscles in thyroid eye diseasePLOS One

Dear Dr. Tagami,

Thank you for submitting your manuscript to PLOS ONE. After careful consideration, we feel that it has merit but does not fully meet PLOS ONE’s publication criteria as it currently stands. Therefore, we invite you to submit a revised version of the manuscript that addresses the points raised during the review process.

We look forward to receiving your revised manuscript.

Kind regards,

Atsushi Asakura, Ph.D

Academic Editor

PLOS One

Journal Requirements:

Additional Editor Comments (if provided):

Reviewers' comments:

Reviewer's Responses to Questions

**Comments to the Author**

1. Is the manuscript technically sound, and do the data support the conclusions?

Reviewer #1: Partly

Reviewer #2: Yes

2. Has the statistical analysis been performed appropriately and rigorously? 

Reviewer #1: Yes

Reviewer #2: N/A

3. Have the authors made all data underlying the findings in their manuscript fully available?

Reviewer #1: Yes

Reviewer #2: Yes

4. Is the manuscript presented in an intelligible fashion and written in standard English?

Reviewer #1: Yes

Reviewer #2: Yes

5. Review Comments to the Author

Reviewer #1: Dear Authors,

This is an interesting topic.

However, I have some questions and suggestions:

1. You should mention the MRI parameters and protocols.

2. You should combine STIR images with T1.

3. This afirmation is controverse: This might be attributed to the fact that the boundary between the SRM and the upper levator muscle is frequently ambiguous

4. The signal intensity of the muscles can be informed, and a comparison between normal signal intensity and hyperintensity (suggesting active disease) can be included.

5.INCLUDE THAT

The signal intensity ratio of T1 and T2 images was also promising in indicating disease activity

Kirsch E, Kaim AH, De Oliveira MG, von Arx GG. Correlation of signal intensity ratio on orbital MRI-TIRM and clinical activity score as a possible predictor of therapy response in graves’ orbitopathy-a pilot studyat 1. 5 t. Neuroradiol (2011) 53:S99. doi: 10.1007/s00234-011-0913-8.

Politi LS, Godi C, Cammarata G, Ambrosi A, Iadanza A, Lanzi R, et al. Magnetic resonance imaging with diffusion-weighted imaging in the evaluation of thyroidassociated orbitopathy: getting below the tip of the iceberg. Eur Radiol (2014) 24 (5):1118–26

Luccas R, Riguetto CM, Alves M, Zantut-Wittmann DE, Reis F. Computed tomography and magnetic resonance imaging approaches to Graves' ophthalmopathy: a narrative review. Front Endocrinol (Lausanne). 2024 Jan 8;14:1277961.

These studies pointed out the method used to indicate disease activity in extraocular muscles, helping to determine the best treatment approach, because T1 and T2 images are sequences collected routinely.

Reviewer #2: This is very interesting topic and very relevant unmet need. So far in the literature there are scarce attempts to objectivise MRI imaging in patients with thyroid eye disease . Still there are no broadly accepted method enabling to do so. The role of AI in this area of medicine is well known and should be considered as probably most promising one. Designing the method which enables every clinician, regardless to their experience, would serve and prompt earlier recognition of a patients who need medical attention whether it is in a form of emerging designated drugs and / or surgery.

I would like to thank you for your efforts and congratulate on this straight forward paper.

6. PLOS authors have the option to publish the peer review history of their article (what does this mean?). If published, this will include your full peer review and any attached files.

Reviewer #1: No

Reviewer #2: No

---

## [Author Response · Author response to Decision Letter 1]

15 Apr 2026

Response to Reviewer #1

1. You should mention the MRI parameters and protocols.

Response: We appreciate the reviewer's insightful comment. We have added MRI parameters and protocols to the Methods section.(Line:118-123)

2. You should combine STIR images with T1.

Response: We appreciate the reviewer's insightful comment. You make an excellent point regarding the potential benefits of combining T1 images with STIR sequences for more precise anatomical segmentation. As noted in the limitations of this discussion, the segmentation of STIR images may differ in size from the actual EOM. However, for several strategic reasons, this study intentionally focused on developing an AI model using only STIR sequences. The first reason is to prevent alignment artifacts. MRI examinations cannot acquire images under each condition (T1 and STIR) simultaneously. Orbital tissue is very small, and eye movements and head movements can cause positional shifts between STIR and T1 images. Precise alignment is necessary to combine these, and to prevent errors that occur in this process, we adopted only STIR images, which can measure SIR and volume simultaneously. The second is clinical utility and workflow simplification. Our primary goal was to develop a "lean" and efficient model that can be easily integrated into routine clinical practice. STIR is the most critical sequence for assessing disease activity in TED due to its sensitivity to edema. By demonstrating that high-precision segmentation and SIR (Signal Intensity Ratio) calculation can be achieved using only a single sequence, we offer a more practical tool that reduces the time and computational resources required for image processing. We have revised and added these considerations to the main text. (Line:243-258)

3. This afirmation is controverse: This might be attributed to the fact that the boundary between the SRM and the upper levator muscle is frequently ambiguous

Response: We appreciate the reviewer's insightful comment. We agree with you that the term "ambiguous" may lack anatomical precision. We intended to convey the technical difficulty of separating the SRM from the adjacent levator palpebrae superioris muscle on STIR images, especially in the presence of inflammatory edema. These two muscles are often functionally and radiologically considered as the "Superior Rectus-Levator Palpebrae Superioris complex" because they are tightly adherent and often enlarge simultaneously in TED. The high signal intensity in STIR and the pathological enlargement frequently obliterate the fat plane between them, leading to partial volume effects. We have revised the Discussion to clarify that this anatomical and radiological challenge in segmentation likely influenced the performance and subsequent statistical analysis for the SRM. (Line:235-242)

4. The signal intensity of the muscles can be informed, and a comparison between normal signal intensity and hyperintensity (suggesting active disease) can be included.

Response: We appreciate the reviewer's insightful comment. We agree that providing an objective comparison of signal intensity is crucial for assessing disease activity. As this study focused on the development and validation of an AI model using a retrospective dataset of patients with active TED, it does not currently include a comparison with a healthy control group or an inactive TED group. However, in this study, we addressed this by evaluating the signal intensity of the extraocular muscles through the SIR. By normalizing the signal against an internal reference, we provided a quantitative measure that represents the spectrum of "hyperintensity" observed in active disease. This approach aligns with the methodology described by Kirsch et al. (2010) [10], who demonstrated the SIR's potential as an objective biomarker.

We agree that future research incorporating clinical activity data (For example, CAS) and control groups will be necessary to define definitive SIR thresholds for distinguishing between normal and active inflammation. This study establishes the foundational automated methodology for such large-scale evaluations. We have added this point to the Discussion secssion. (Line:277-288)

5.INCLUDE THAT、、

Response: We appreciate the reviewer's insightful comment. We have added the references you provided to the introduction. (Line:90-94, 99-103)

Response to Reviewer #2

This is very interesting topic and very relevant unmet need. [...] I would like to thank you for your efforts and congratulate on this straight forward paper.

Response: We are sincerely grateful to the reviewer for the highly encouraging comments and for recognizing the clinical significance of our work. We agree that there is a substantial unmet need for objective MRI evaluation in thyroid eye disease, and we are pleased that the reviewer found our AI-based approach to be a promising step toward addressing this challenge. We appreciate the time and effort invested in reviewing our manuscript.

---

## [Editor Report · Decision Letter 1]

23 Apr 2026

Development and evaluation of AI model with deep learning for segmentation of extraocular muscles in thyroid eye disease

PONE-D-26-06489R1

Dear Dr. Tagami,

We’re pleased to inform you that your manuscript has been judged scientifically suitable for publication and will be formally accepted for publication once it meets all outstanding technical requirements.

Kind regards,

Atsushi Asakura, Ph.D

Academic Editor

PLOS One
---

## [Editor Report · Acceptance letter]

PONE-D-26-06489R1

PLOS One

Dear Dr. Tagami,

I'm pleased to inform you that your manuscript has been deemed suitable for publication in PLOS One. Congratulations! Your manuscript is now being handed over to our production team.

Kind regards,

on behalf of

Dr. Atsushi Asakura

Academic Editor

PLOS One